# Dietary Components Associated with the Risk of Gastric Cancer in the Latin American Population: A Systematic Review and Meta-Analysis

**DOI:** 10.3390/foods14061052

**Published:** 2025-03-19

**Authors:** Azalia Avila-Nava, Ana Ligia Gutiérrez-Solis, Oscar Daniel Pacheco-Can, Ian Yeshua Sagols-Tanoira, Roberto González-Marenco, Ana Gabriela Cabrera-Lizarraga, Jesús Abraham Castillo-Avila, Miguel Alberto Aguilar-Franco, Rodolfo Chim-Aké, Héctor Rubio-Zapata, Mariela Reyes-Sosa, Isabel Medina-Vera, Martha Guevara-Cruz, Katy Sánchez-Pozos, Roberto Lugo

**Affiliations:** 1Unidad de Investigación, Hospital Regional de Alta Especialidad de la Península de Yucatán, IMSS-Bienestar, Mérida 97130, Mexico; zomi33@gmail.com (A.A.-N.); ganaligia@gmail.com (A.L.G.-S.); oscar_dpc@outlook.com (O.D.P.-C.); rob_marenco@hotmail.com (R.G.-M.); rodolfochim@hotmail.com (R.C.-A.); 2Becario de la Dirección General de Calidad y Educación en Salud (DGCES), Secretaría de Salud, Ciudad de México 11400, Mexico; 3Escuela de Medicina, Universidad Marista de Mérida, Mérida 97302, Mexico; sagolsian@gmail.com (I.Y.S.-T.); anagabycabrera_liza@icloud.com (A.G.C.-L.); aguilarmiguel075@gmail.com (M.A.A.-F.); 4Facultad de Medicina, Universidad Autónoma de Yucatán, Mérida 97000, Mexico; hector.rubio@correo.uady.mx; 5Facultad de Medicina, Universidad Autónoma de Campeche, Campeche 24039, Mexico; abraham_castillo06@hotmail.com; 6Secretaría de Ciencia Humanidades, Tecnología e Innovación, Universidad Autónoma de Yucatán, Mérida 97000, Mexico; mreyes@secihti.mx; 7Departamento de Metodología de la Investigación, Instituto Nacional de Pediatría, Ciudad de México 04530, Mexico; isabelj.medinav@gmail.com; 8Fisiología de la Nutrición, Instituto Nacional de Ciencias Médicas y Nutrición Salvador Zubirán, Ciudad de México 14080, Mexico; marthaguevara8@yahoo.com.mx; 9División de Investigación, Hospital Juárez de México, Ciudad de México 07760, Mexico; katypozos@gmail.com

**Keywords:** gastric cancer, dietary components, food, macronutrients, micronutrients

## Abstract

Gastric cancer is one of the most frequent cancer types in the Latin American population, and its development is related to multiple factors, including diet. The present systematic review and meta-analysis, based on PRISMA, aims to determine dietary components associated with gastric cancer in Latin America. The dietary components were divided into food and micro- and macronutrients. Meta-analyses were performed for the different groups of foods, and the effects were calculated using Odds Ratios. A total of 483 studies were identified; thirteen articles were included after removing duplicates and applying the inclusion and exclusion criteria. The meta-analysis for the different groups of food showed that daily consumption of vegetables (OR 0.54; 95% CI [0.41, 071]) and fruits (OR 0.57; 95% CI [0.45, 0.73]) were protective factors for gastric cancer; consumption of fresh meat and eggs (OR 1.47; 95% CI [1.12, 1.95]), and salted, canned, and pickled foods (OR 2.30; 95% CI [1.10, 4.80]) were risk factors for gastric cancer. Consuming carotenoids, plant sterols, total polyphenols, proteins, and polyunsaturated fats was a protective factor regarding micro- and macronutrients. Therefore, the consumption of nitrite and nitrate in meat products and capsaicin was considered a risk factor for gastric cancer.

## 1. Introduction

Latin American countries are among the 20 countries with the highest incidence of gastric cancer. It is estimated that the number of cases will continue to increase, especially in high mountainous regions [1]. In Latin America and the Caribbean, gastric cancer ranks sixth in terms of incidence with 8.5 per 100,000 inhabits, and also sixth in mortality with 6.5 per 100,000 inhabitants according to the age-standardized ratio of the International Agency for Research on Cancer (GLOBOCAN, 2022) [2]. Gastric cancer mortality in Latin America is not homogeneous; for example, countries such as Costa Rica, Guatemala, and Chile have high mortality rates, while Mexico and Cuba have average (but increasing) mortality rates [3].

Several risk factors have been associated with gastric cancer; among these is lifestyle, which includes diet. Diet in Latin America is a factor of great relevance, and it depends on several factors and one of the most important is social inequality. This inequality is related to insufficient access to safe and quality food, as well as to higher levels of food insecurity and malnutrition. This scenario has been aggravated by the changes in diet, which have also occurred in these countries [4]. Latin America has undergone a nutritional transition from traditional foods such as whole plant foods with minimal processing or refinement of carbohydrates to diets that include animal products, ultra-processed foods constituted by refined carbohydrates, high sodium, saturated fats, and sugar-sweetened beverages [5]. This has promoted an increase in the consumption of foods that contain compounds that have been associated with gastric cancer, mainly present in processed foods of animal origin [6], and a decrease in foods of vegetable origin, which have bioactive compounds that could reduce its development and/or progression [7]. In fact, it has been reported that the consumption of red meat, processed meat, and diets high in salt have a strong association with gastric cancer [1]. This is of the utmost importance, since it is a modifiable factor; so knowing what type of food or nutrients are associated with this pathology is important. Thus, the present review aimed to determine the main dietary components associated with gastric cancer present in the Latin American population.

## 2. Materials and Methods

### 2.1. Study Design and Search Strategy/Data Source

A systematic review was performed independently by five researchers (ODPC, MAAF, AGCL, IST, and JACA) using published articles on the dietary habits of subjects with gastric cancer in Latin American countries. The studies included were identified by searching in English and Spanish using online sources and databases such as MEDLINE/PubMed, Web of Science, Cochrane Library, and LILACS library. Gray literature was searched using Google Scholar and the Library of the National Autonomous University of Mexico (UNAM). The combination of the keywords that yielded the best results in databases was ([Dietary] AND [Gastric cancer] AND [South America]); ([Dietary] AND [Gastric cancer] AND [Mexico]). The search was performed for titles and/or abstracts and was carried out between December 2022 and January 2023. The systematic review has been registered in the International Prospective Register of Systematic Reviews (PROSPERO) with the number CDR42023442460 for evaluation.

### 2.2. Selection of Studies

After removing duplicates, the same authors independently screened the titles and abstracts for eligibility evaluation, based on the inclusion criteria. The inclusion criteria for this study were (I) subjects with a diagnosis of primary gastric cancer; (II) subjects over 18 years of age; (III) Latin American population; (V) studies that used a standardized instrument for assessing the dietary components in case–control participants; and (VI) articles published in 2000 or later were included. The exclusion criteria applied were (I) manuscripts written in languages other than Spanish or English; (II) studies that were systematic reviews and meta-analyses; (III) studies with animal models or cell lines; (IV) abstracts, congress abstracts, case reports, case series, and incomplete or unavailable documents; (V) studies that did not include dietary evaluation of the population; and (VI) studies that did not include gastric cancer as a primary tumor. All abstracts were evaluated for eligibility (in some cases, studies were evaluated in full-length) based on the established criteria, and there was no recruitment period or sample size restriction.

### 2.3. Data Extraction

The present study was based on the Preferred Reporting Items for Systematic Reviews and Meta-Analysis (PRISMA) criteria [8]. Data from all the selected articles were independently extracted by three researchers (AAN, ALGS, and RL) into a predefined database, including the first author, year of publication, the title of the study, country, study design, population (number and percentage for cases and control), and the foods or macro- and micronutrients that showed the risk or protective effect (odd ratios from the second quartile) reported in all studies. The Prisma checklist is available in Appendix A.

### 2.4. Study Quality Assessment

The quality of the methodology and the risk of bias for each study were evaluated using the Critical Appraisal tools in JBI Systematic Reviews [9]. In case of discrepancy among researchers (ODPC, MAAF, AGCL, IST, and JACA), the final consensus was reached by another investigator (AAN, ALGS, and RL). The checklist for case–control studies was used to assess the articles included in this review. The quality of comparability of the groups, parity, exposure measures, validity of exposure and results, and confounding factors were analyzed.

### 2.5. Statistical Analysis

Meta-analyses were performed for different groups of foods (vegetables, fruits, fresh meat, and egg; processed meat, cereal, and tubers; fish and seafood; and salted, canned, and pickled foods) using a random effects model and the odds ratio (OR) method. The number of cases and controls of the second quartiles were used to determine the potential protective effect or risk of each group of foods. Forest plots were performed according to the consumption of the different groups of food for cases and controls with respect to the total number of subjects recruited in each study and group (N). The OR and 95% interval confidence (95% IC) were calculated for each study and the pooled group using Review Manager (RevMan) statistical software version 5.4 (The Cochrane Collaboration). Statistical heterogeneity was assessed using the *I*^2^ index. Values of *p* < 0.05 were considered statistically significant.

## 3. Results

A total of 483 studies were identified. After removing duplicates, 321 studies were considered as candidates for this review. Altogether, 162 articles were eliminated due to duplications in the search, and their titles and abstracts were reviewed. After applying the inclusion and exclusion criteria, 13 articles were included in this systematic review and meta-analysis (Figure 1).

### 3.1. Characteristics of Studies

The publications implicate a period from 2000 to 2022 that involves a total of 8742 subjects (2644 cases and 6098 controls). Among the thirteen studies included, five were from Uruguay, four were from Mexico, two were from Brazil, one was from Venezuela, and one from Colombia.

The dietary components were divided into food and micro- and macronutrients. The studies performed by De Stefani et al., (2001) [10], De Stefani et al., (2001) [11], Muñoz et al., (2001) [12], Nishimoto et al., (2002) [13], De Stefani et al., (2004) [14], Campos et al., (2006) [15], Delgado-Figueroa et al., (2017) [16], Trujillo-Rivera et al., (2018) [17], and Peres et al., (2022) [18] were grouped because they analyzed different food groups (Table 1).

The studies performed by De Stefani et al., (2000) [19], De Stefani et al., (2000) [20], Muñoz et al., (2001) [12], López-Carrillo et al., (2003) [21], Hernández-Ramírez et al., (2009) [22], and Trujillo-Rivera et al., (2018) [17] analyzed the main micronutrients and macronutrients associated in the diet from subjects with gastric cancer (Table 2). On the other hand, the studies conducted by Muñoz et al., (2001) [12], and Trujillo-Rivera et al., (2018) [17] analyzed dietary components, the type of food, and micro- and macronutrients.

The meta-analysis grouped food according to specific characteristics, such as all vegetables, fruits, fresh meat and eggs, processed meat, cereals and tubers, fish and seafood, and salted or canned food.

### 3.2. Vegetables

To determine whether the intake of vegetables can be a protective or a risk factor in subjects with gastric cancer, we pooled six articles that assess the effect of vegetables. The study concluded by De Stefani (2001) [11] showed that all vegetables were at protective risk in subjects with gastric cancer from Uruguay. Nishimoto et al., (2002) [13] assessed the effect of green vegetables, and other vegetables, considering the last as a protective factor in subjects with gastric cancer in the Brazilian population; Campos et al., (2006) [15], evaluate the impact of all vegetables as a protective factor in the Colombian population; Delgado-Figueroa et al., (2017) [16] show the positive effect in the consumption of green vegetables in subjects with diffuse and intestinal gastric cancer from Mexican population; and finally, the study by Trujillo-Rivera et al., (2018) [17] assesses the effect of all vegetables in subjects with gastric cancer from Mexico. The overall effect on this food group showed that the consumption of vegetables is considered a protective factor in the population of Latin America (OR 0.54; 95% CI [0.41–071]; *p* < 0.0001). Also, the results showed low heterogeneity (*I*^2^ = 18%) between studies (Figure 2).

### 3.3. Fruits

To assess whether the intake of fruits is considered a protective or risk factor in subjects with gastric cancer, we pooled five articles that evaluated the fruit consumption effect on the risk of gastric cancer. The study by De Stefani et al., (2001) [11] assessed all fruits in subjects from Uruguay, showing a protective effect against gastric cancer. In addition, the studies conducted by Nishimoto et al., (2002) [13] and Campos et al., (2006) [15] assessed the effect of all types of fruit on populations from Brazil and Colombia, respectively; however, the studies did not show any effect. Similar results were reported by Delgado-Figueroa et al., (2017) [16] when evaluating the citrus fruit and other fruits in subjects with diffuse and intestinal gastric cancer in the Mexican population. Nevertheless, Trujillo-Rivera et al., (2018) [17] reported that the consumption of any kind of fruit in the Mexican population was a protective factor. The analysis of the pooled group showed a favorable effect on fruit consumption in subjects with gastric cancer (OR 0.57; 95% CI [0.45–0.73]; *p* < 0.001) being a protective factor. In addition, the results showed no heterogeneity in the studies (*I*^2^ = 0%) (Figure 3).

### 3.4. Fresh Meat and Eggs

Three articles that evaluated the consumption of fresh meat and eggs as risk factors for subjects with and without gastric cancer were pooled. Studies by De Stefani et al., (2001) [10] and Delgado-Figueroa et al., (2017) [16] reported that the consumption of red and pork meat were risk factors for developing gastric cancer in populations from Uruguay and Mexico, respectively. Moreover, the study by Nishimoto et al., (2002) [13] showed that daily egg consumption is also considered a risk for gastric cancer in Brazil. The forest plot for this group showed that the consumption of fresh meat and eggs is considered a risk factor in the presence of gastric cancer for the Latin American population (OR 1.47; 95% CI [1.12–1.95]; *p* = 0.006) (Figure 4). In addition, the pooled group showed low heterogeneity (*I*^2^ = 38%).

### 3.5. Processed Meat

Only two articles evaluated processed meat in populations from Uruguay and Mexico. In the study by De Stefani et al., (2001) [10], they found that only processed meat was a risk factor (OR 1.6; 95% CI 1.08–2.59) in the population of Uruguay, whereas in the study by Delgado-Figueroa et al., (2017) [16] for Mexican population, they did not find an association between the consumption of sausage, ham, bacon, and general processed meat with gastric cancer. The overall effect observed in this group was not significant (OR 1.03; 95% CI [0.65–1.65]; *p* = 0.89), showing moderate heterogeneity (*I*^2^ = 39%) in the analysis (Appendix A).

### 3.6. Cereals and Tubers

The studies performed by Nishimoto et al., (2002) [13], De Stefani et al., (2004) [14], and Delgado-Figueroa et al., (2017) [16] evaluated tubers, rice and beans, and potatoes in subjects from Uruguay, Brazil, and Mexico. The forest plot does not show an effect (protective or risk) of the consumption of cereals and tubers on subjects with and without gastric cancer in their populations (OR 1.10; 95% CI 0.82–1.47; *p* = 0.53) and shows no heterogeneity in the studies (*I*^2^ = 0%) (Appendix A).

### 3.7. Fish and Seafood

No overall effect of fish and seafood consumption (OR 0.94; 95% IC [0.65–1.37]; *p* = 0.75) was observed after analyzing the group conformed by De Stefani et al., (2001) [10], Nishimoto et al., (2002) [13], and Delgado-Figueroa et al., (2017) [16], showing low heterogeneity (*I*^2^ = 12%) between the studies (Appendix A).

### 3.8. Salted, Canned, and Pickled

Three articles assessed the effect of salted, canned, and pickled foods as protective or risk factors for gastric cancer. Campos et al., (2006) [15] reported that salting meals confer a high risk for gastric cancer in the Colombian population. Delgado-Figueroa et al., (2017) [16] and Trujillo-Rivera et al., (2018) [17] observed that sardines and pickled foods were risk factors in the Mexican population. The overall effect of the consumption of products of this group was a risk factor for gastric cancer (OR 2.30; 95% CI [1.10–4.80]; *p* < 0.03). In addition, the group showed moderate heterogeneity in the studies (*I*^2^ = 44%) (Figure 5).

As mentioned above, the dietary components associated with gastric cancer can be divided into food and micro- and macronutrients. Only six articles that assessed the micronutrients and macronutrients as protective or risk factors in subjects with gastric cancer for the Latin American population were selected (Table 2).

### 3.9. Micronutrients

Through a similar analysis performed for the types of food, micronutrients can behave as a protective or risk factor associated with cancer gastric. The two studies conducted by De Stefani et al., in 2000 [19,20] reported that carotenoids (a-carotene and lycopene), vitamin C, and plant sterols (b-sitosterol and total phytosterols) were protective in the population from Uruguay. Moreover, in another study in the same population, De Stefany et al., (2001) [10] reported that the consumption of nitroso dimethylamine and methionine were a risk factor for gastric cancer. In Venezuela, niacin was considered a protective factor in this population according to a study by Muñoz et al., (2001) [12]. Elsewhere, the studies conducted by López-Carrillo et al., (2003) [21] and Trujillo-Rivera et al., (2018) [17] in the Mexican population showed that the regular consumption of capsaicin (30.0–89.9 mg/d) is considered a risk factor for gastric cancer. Finally, the study by Hernández-Ramírez et al., (2009) [22] concluded that the consumption of polyphenols (cinnamic acids, lignans, and coumestrol) is considered a protective factor for gastric cancer. In addition, the nitrate in fruits and vegetables also are considered a protective factor in the Mexican population. However, the consumption of nitrite and nitrate from processed meat products was considered a risk factor.

### 3.10. Macronutrients

Only one article published by Muñoz et al., (2001) [12] showed the effect of the macronutrients in subjects with gastric cancer. They concluded that the consumption of protein and polyunsaturated fat decreased the risk of gastric cancer in the population from Venezuela.

The quality assessment showed that all studies had comparable case–control groups with the presence of gastric cancer being the only difference; the same validated and reliable instruments were applied in both cases and control groups. Outcomes were assessed in a validated and reliable manner, the period of exposure to potential protective and risk factors was sufficient, and appropriate statistical analysis was used. The main risks of bias were the lack of identification and strategies to address confounders, lack of the same criteria for cases and controls identification, and lack of appropriate matching (Appendix A).

## 4. Discussion

Gastric cancer is one of the most common types of cancer worldwide. Generally, gastric cancer has a poor prognosis. The first line of treatment is surgical resection followed by adjuvant therapies [23]. However, in advanced stages, surgical resection has not shown improved survival in subjects with gastric cancer [24].

Gastric cancer is associated with multiple factors, such as smoking, alcohol intake, obesity, the presence of Epstein–Barr virus, the presence of *Helicobacter pylori*, family history of cancer, and diet [25]. Therefore, identifying the protective and risk factors associated with this disease should be a mandatory task for health authorities.

Although eating habits are diverse in several parts of Latin America, we identify groups of foods that are similar between countries. In the studies conducted by De Stefani et al., for the Uruguayan population, they concluded that the consumption of all types of fruits (>195.9 g/day) and vegetables (>29.5 g/day) are enough to reduce the risk factor for gastric cancer [11]. Meanwhile, the regular consumption of grains, tubers, and carbohydrates, or the consumption of red (>152.5 g/day) and processed meat (>29.6 g/day) increases the risk for gastric cancer [10,14]. Interestingly, Uruguay and Argentina rank first and second position in the consumption of beef in the world, respectively, with about 60 Kg of meat per year per capita [26], conferring a high probability for the development of gastric cancer. In fact, according to the International Agency for Research on Cancer, Uruguay ranks seventh in gastric cancer incidence (9.2 per 100,000 inhabitants) and twelfth in mortality (6.1 per 100,000 inhabitants) [27]. Furthermore, the Uruguayan population has high levels of overweight and obesity (65% of the population) in subjects aged 25 to 64, which contributes to a major problem not only because its treatment as a chronic non-communicable disease is not contemplated by its national health system [28], but also because these comorbidities increase the risk for cancer.

Nishimoto et al., concluded that the daily consumption of fruits and vegetables within the Brazilian population reduces the risk for gastric cancer [13]. Conversely, they concluded that the daily consumption of eggs increases the risk of gastric cancer. This is opposed by De Stefani et al. [14] when concluding that eggs are a protective factor for cancer. Perhaps the discrepancy in results is due to the frequency of egg consumption because there is evidence that the increase in the consumption of some foods can promote the presence of some pathologies including gastric cancer [29]. Among the factors associated with gastric cancer is the high consumption of salt or salted foods. Excessive salt intake can destroy the stomach mucosa barrier, causing inflammation and damage to the mucosa, atrophic gastritis, and decreased gastric acid, leading to the development of *H. pylori* infection [29,30]. In this context, the study by Peres et al., concludes that the consumption of salted bread, processed meat, and processed and ultra-processed food in São Paulo (>1448 g/day) and Belém (Amazon region) (>913 g/day), Brazil, increases the risk of gastric cancer [18]. Similar results were reported by Campos et al., when analyzing salted meals in the study conducted in Colombia [15]. In this context, processed meat contains nitrosamines that are formed from nitrates and nitrites added during the curing, smoking, dehydrating, and salting process [31]. The role of nitrosamines is still controversial as some studies describe potential carcinogenesis through the endogenous nitrosation forming N-nitroso compounds [32,33], which are present in the environment such as water, cigars, air or food as main exponents of processed foods and smoked products. These nitrosamine compounds in the presence of salt can promote carcinogenesis increasing the probability of endogenous mutations in the stomach [34]. Hence, the consumption of salted and processed meat in Colombia can explain the increase in the prevalence of gastric cancer (12.9 per 100,000 inhabitants) being the highest incidence in Latin America, only surpassed by Peru, Chile, and Costa Rica [27]. It is important to note that the risk of cancer from nitrate consumption depends primarily on the sources from which these compounds are ingested, since nitrates are present in a wide variety of foods. In fact, between 80% and 85% of human exposure to nitrates comes from vegetables [35]; however, these foods are not linked to the development of gastric cancer because vegetables are also rich in vitamin C, vitamin E, polyphenols and fiber, and compounds that reduce the formation of nitrosamines, which are identified as carcinogens [36]. On the other hand, processed meats lack reducing agents, allowing absorbed nitrate to be actively transported to the salivary glands, where oral bacteria reduce it to nitrite [37]. In the digestive tract, nitrites can be converted into nitrosating agents, which can lead to the nitrosation of biogenic amines and, consequently, the formation of N-nitrosamines [37,38].

In Mexico, the consumption of pork meat, canned sardines, and pickled food was a risk factor for gastric cancer [16,17]. It is important to note that the Mexican population has a traditional diet with a high ingestion of fat. In addition, the population showed overweight (38.3%) and obesity (36.9%) in persons over 20 years, non-communicable diseases, sedentary life, and generally bad eating habits, all of which increase the probability of gastric cancer in this population [39,40,41]. Our analysis confirms that both pooled salting, canned and pickled foods, and the pooled consumption of fresh meat and eggs, showed risk factors for gastric cancer. Also, the consumption of fruit (>1 piece/day) and vegetables (>1 piece/day) were protective factors for gastric cancer.

In this study, we also analyzed the micro- and macronutrients found in foods. The presence of antioxidants identified in fruits and vegetables were a protective factor for gastric cancer. In this case, the absorption of vitamin C, a-carotene, and lycopene act synergistically to reduce oxidative stress on the gastric mucosa and the luminal surface, modulating the kinetics of cell growth, preventing carcinogenesis, and inhibiting the formation of N-nitroso compound in the stomach [42]. Moreover, the consumption of polyphenols and other phytochemicals can decrease the formation, bioactivation, and carcinogenicity of heterocycle amine and nitrosamines [42,43]. Polyphenols can inhibit the formation of lipid peroxidation products in the stomach when ingesting red meat. Polyphenols have a greater capacity to eliminate reactive oxygen species than vitamin C, vitamin E, and carotenoids and this explains their protective role against gastric cancer [44]. A diet rich in polyphenols and vitamin C, like the Mediterranean diet, helps to maintain redox homeostasis in the stomach during meals [42]. In the analysis of our results, we observed that protein was a protective macronutrient for gastric cancer, but the forest plot showed that fresh meat was a risk factor. This difference could be explained because in the study by Muñoz et al., (2001) [12], no distinction was made between protein of animal and vegetable origin. Neither was the degree of processing of said protein evaluated. In addition, in this study the amount of protein intake was not reported, this being an important variable.

The consumption of capsaicin is still controversial because the article published by Muñoz et al. [12] performed in the Venezuelan population describes that the consumption of chili (<once/week) reduces the risk for gastric cancer. However, chili and capsaicin are one of the main products in Mexican gastronomy. Many products contain this compound, ranging from candies to sauces, or entire chili-based dishes. Maybe for this reason, in the Mexican population the consumption of >29.9 mg/day of capsaicin is considered a risk factor for gastric cancer [17]. The risk is not only for the quantity and frequency of capsaicin consumption but also due to the presence of several factors. The study conducted by Lopez-Carrillo describes that the consumption of chili pepper is high and there are some associations between the moderate consumption of capsaicin (chili) and the risk of gastric cancer in genetically susceptible individuals (carriers of alleles IL1B-31C > T) and infected by the most virulent *H. pylori* (Cag A positive) [45]. It has also been demonstrated that this molecule promotes metastasis, particularly in lung and gastric cancers, due to its agonist activity on the Transient Receptor Potential Vanilloid-1 (TRPV1), a non-selective ligand-gated cation channel with high permeability to ionic calcium (Ca^2+^). TRPV1 is abnormally expressed in gastric tumor cells, leading to the polymerization of cytoskeletal proteins, which, in turn, provides these cells with increased motility [46]. In addition, the presence of non-communicable diseases is liked to chronic low-grade inflammation, both of which have been associated with the increased risk of cancer; in this sense, the Mexican population presented all of these associated factors, increasing the probability of the risk of gastric cancer [40,41,47].

This study presents the following strengths and limitations: firstly, this study showed the dietary components present in the diet of the population of Latin America. In addition, we analyzed the association of the main groups of food and the micro- and macronutrients with gastric cancer by population. The limitations in the studies included the fact that not all investigations describe the main characteristics of their population; the sample size varied between studies; not all studies describe the quantity and frequency of the food or micro- and macronutrient intake; the instruments used for the measurements are not equivalents in all of the studies showing discrepancies in the results; and finally, the results were different because the dietary habits are different between Latin American countries. In addition, we recommend cohort studies in which the diagnosis of cases and controls is confirmed by biopsy analysis. Furthermore, the use of validated instruments to collect information on the frequency, quantity, storage, type of preparation, cooking method, and time of consumption of the main foods in each region is needed.

## 5. Conclusions

This systematic review and meta-analysis determined that the consumption of daily fresh fruits and vegetables contributed to the decrease in gastric cancer. In addition, the regular consumption of fresh meat and eggs, and salted, canned, and pickled food was deemed a risk factor for gastric cancer. Moreover, the regular consumption of micro- and macronutrients such as carotenoids, sterols, polyphenols, proteins, and polyunsaturated fatty acids showed a protective effect against gastric cancer, and the regular consumption of capsaicin can increase the risk for gastric cancer in the Latin American population.

## Figures and Tables

**Figure 1 foods-14-01052-f001:**
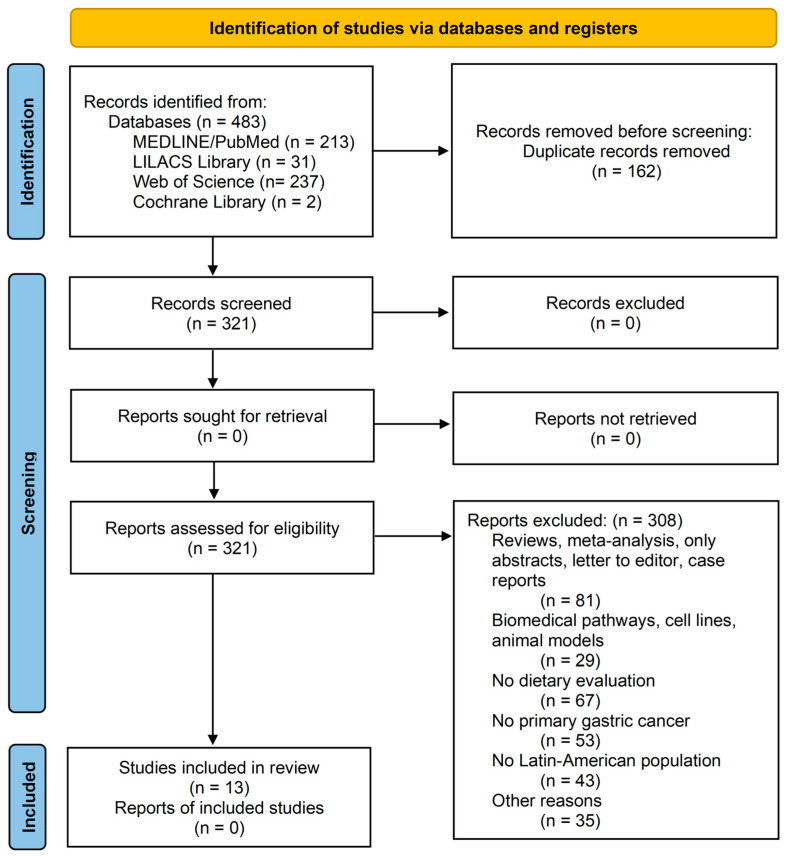
Prisma flow diagram.

**Figure 2 foods-14-01052-f002:**
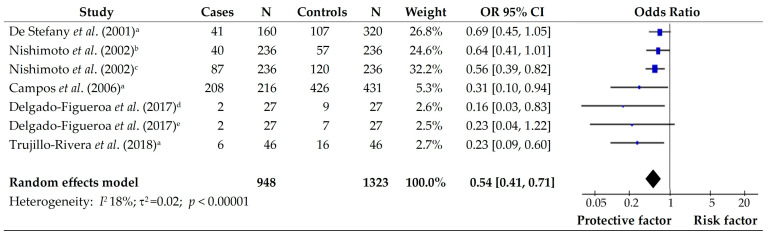
Forest plot of the OR for the consumption of vegetables in subjects with and without gastric cancer. a: all vegetables; b: green vegetables; c: other vegetables; d: green vegetables in diffuse gastric cancer; e: green vegetables in intestinal gastric cancer [11,13,15,16,17].

**Figure 3 foods-14-01052-f003:**
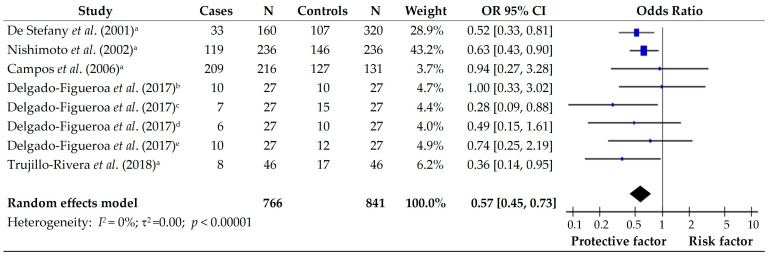
Forest plot of the OR for the consumption of fruits in subjects with and without gastric cancer. a: all fruits; b: citrus fruits in diffuse gastric cancer; c: other fruits in diffuse gastric cancer; d: citrus fruits in intestinal gastric cancer; e: other fruits in intestinal gastric cancer [11,13,15,16,17].

**Figure 4 foods-14-01052-f004:**
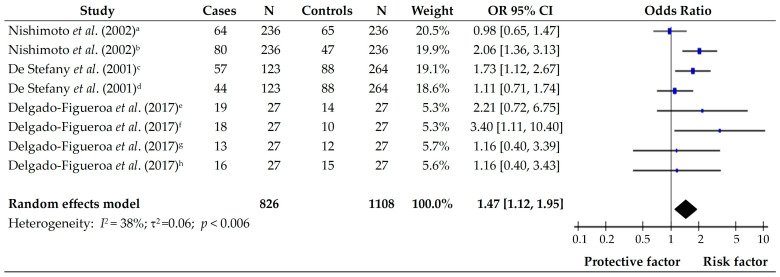
Forest plot of the OR for the consumption of fresh meat and eggs in subjects with and without gastric cancer. a: beef; b: egg; c: all red meat; d: all white meat; c: beef; d: egg; e: beef in diffuse gastric cancer; f: pork in diffuse gastric cancer; g: beef in intestinal gastric cancer; h: pork in intestinal gastric cancer [10,13,16].

**Figure 5 foods-14-01052-f005:**
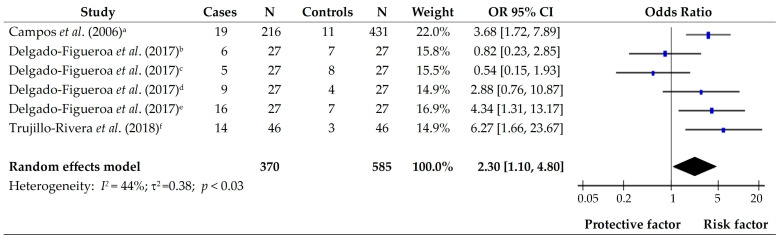
Forest plot of the OR for the consumption of salting, canned, and pickled foods in subjects with and without gastric cancer. a: salting meals; b: tuna in diffuse gastric cancer; c: tuna in intestinal gastric cancer; d: sardine in diffuse gastric cancer; e: sardine in intestinal gastric cancer; f: pickled foods [15,16,17].

**Table 1 foods-14-01052-t001:** Consumption of different types of foods related to gastric cancer.

Num	Author/Year/Study	Country	Study Design	Population, n (%)	Cases vs. Control OR [95% CI]	
					Protective Factor	Risk Factor
1	De Stefani et al., (2001) [10]. Meat Consumption and Risk of Stomach Cancer in Uruguay: A Case–Control Study	Uruguay	Case–Control	Cases Males, 91 (74) Females, 32 (26) Control Males, 173 (65.5) Females, 91(34.5)	NR	Consumption of Red meat (>152.5 g/d) 2.4 [1.3–4.4] Processed meat (>29.6 g/d) 2.3 [1.3–4.2] Total meat (164.4–219.8 g/d) 2.6 [1.3–5.2]
2	De Stefani et al., (2001) [10]. Plant foods and risk of gastric cancer: a case–control study in Uruguay	Uruguay	Case- Control	Cases Males, 114 (71.3) Females, 46 (28.8) Control Males, 225 (70.3) Females, 95 (29.7)	Consumption of Raw vegetables (>29.5 g/d) 0.52 [0.31–0.86] Allium vegetables (>21.0 g/d) 0.46 [0.29–0.76] All fruits (>195.9 g/d) 0.33 [0.20–0.56] Citrus fruit (>47.5 g/d) 0.51 [0.31–0.85] Other fruits (>150.2 g/d) 0.34 [0.20–0.57] All vegetables and fruits (>321.1 g/d) 0.33 [0.19–0.55] Plant foods (>427.9 g/d) 0.31 [0.18–0.54]	NR
**3**	Muñoz et al., (2001) [12]. A case–control study of gastric cancer in Venezuela	Venezuela	Case- Control	Cases 292 (37.5) Control 485 (62.5)	Consumption of Chili (<once/w) 0.50 [0.30–0.90] Garlic (every day) 0.50 [0.30–0.70] Onion (every day) 0.40 [0.20–0.70]	NR
**4**	Nishimoto et al., (2002) [13]. Risk factors for Stomach Cancer in Brazil (I): a Case–control study among Non-Japanese Brazilian in São Paulo	Brazil	Case- Control	Cases Males, 170 (72.0) Females, 66 (28.0) Control Males, 170 (72.0) Females, 66 (28.0)	Consumption of Fruit (daily) 0.4 [0.2–0.8] Yellow vegetables (daily) 0.4 [0.2–0.8] Other vegetables (daily) 0.4 [0.2–0.8]	Consumption of Egg (daily) 2.7 [1.5–4.9]
**5**	De Stefani et al., (2004) [14]. Dietary patterns and risk of gastric cancer: a case–control study in Uruguay	Uruguay	Case- Control	Cases Males, 168 (70) Females, 72 (30) Control Males, 672 (70) Females, 288 (30)	Consumption of *^+^ Egg 0.67 [0.47–0.97] Other fruits 0.46 [0.31–0.71] Total fruits 0.51 [0.35–0.74] Total vegetables and fruits 0.54 [0.41–0.85]	Consumption of *^+^ Total grains 1.63 [1.07–2.46] Starchy food 1.55 [1.01–2.38] All tubers 1.69 [1.13–2.52]
6	Campos et al., (2006) [15]. Risk factors of gastric cancer specific for tumor location and histology in Cali, Colombia	Colombia	Case- Control	Cases 216 (33.3) Control 431 (66.7)	Consumption of ^+^ Fruits intake 0.3 [0.1–1.0] Vegetable intake 0.3 [0.1–1.0]	Consumption of ^+^ Salting meals 3.5 [1.6–7.3] Frying foods 1.9 [1.0–3.6]
7	Delgado-Figueroa et al., (2017) [16]. Risk factors associated with diffuse gastric cancer and intestinal histological patterns in adult population from Western Mexico	Mexico	Case- Control	Cases Diffuse gastric cancer 27 (50.9) Intestinal gastric cancer 26 (40.1) Control Diffuse gastric cancer 27 (50.9) Intestinal gastric cancer 26 (40.1)	Diffuse gastric cancer: Consumption of Fruit (>1/d) 0.28 [0.08–0.88] Green vegetables (>1/d) 0.16 [0.03–0.83]	Diffuse gastric cancer: Consumption of Pork meat (>1/w) 3.4 [1.11–10.40] Intestinal gastric cancer: Consumption of Canned sardine (>1/mo) 4.07 [1.25–13.24]
8	Trujillo-Rivera et al., (2018) [17]. Risk factors associated with gastric cancer in Mexico: education, breakfast, and chili	Mexico	Case- Control	Cases Males, 27 (58.7) Females, 19 (41.3) Control Males, 27 (58.7) Females, 19 (41.3)	Consumption of Fresh fruits (≥1 piece/d) 0.31 [0.09–0.89] Fresh vegetables (≥1 piece/d) 0.25 [0.06–0.78]	Consumption of Pickled food (>32.42 g/d) 4.7 [1.30–25.33]
9	Peres et al., (2022) [18]. Consumption of processed and ultra-processed foods by patients with stomach adenocarcinoma: a multicentric case–control study in the Amazon and southeast region of Brazil	Brazil São Paulo (Southeast region)	Case- Control	Cases Males, 214 (63.1) Females, 79 (36.9) Control Males, 315 (57.1) Females, 236 (42.9)	Consumption of Whole grain bread (>6.7 g/d) 0.62 [0.43–0.90] Pasta (>32 g/d) 0.53 [0.35–0.80]	Consumption of Salted bread (> 44 g/d) 2.45 [1.63–3.67] Leguminous (beans and lentils) (>102 g/d) 1.55 [1.02–2.34] French fries (cassava fries) (>5 g/d) 1.96 [1.31–2.94] Fried and roasted meats (>185 g/d) 1.97 [1.27–3.07] Processed meat, sausage, cold cuts and ‘tropeiro’ beans (>44 g/d) 2.98 [1.93–4.59] Pizza, fried and baked snacks, popcorn, and snack (>37 g/d) 1.91 [1.27–2.87] Processed and Ultra-processed food (>1448 g/d) 2.11 [1.39–3.22]
		Brazil Belém (Amazon region)	Case- Control	Cases Males, 86 (61.4) Females, 54 (38.6) Control Males, 86 (61.4) Females, 54 (38.6)	Consumption of Whole grain bread (>1.7 g/d) 0.08 [0.02–0.36] Pasta (>54 g/d) 0.48 [0.26–0.87] French fries (cassava fries) (>5 g/d) 0.29 [0.17–0.50] Pizza, fried and baked snacks, popcorn, and snack (>27.6 g/d) 0.29 [0.16–0.53]	Consumption of Salted bread (>101 g/d) 4.44 [2.33–8.47] White rice (>189 g/d) 11.85 [6.05–23.21] Leguminous (beans and lentils) (>66.6 g/d) 12.14 [5.76–25.58] Fried and roasted meats (>166 g/d) 2.45 [1.36–4.42] Processed and Ultra-processed food (>913 g/d) 13.21 [6.56–26.62]

OR, odds ratio; 95% CI, 95% confidence interval; NR, not reported; g, grams; d, day; w, week; mo, month; * OR from second tercile (Q2); ^+^ Consumption or frequency is not reported.

**Table 2 foods-14-01052-t002:** Consumption of micro- and macronutrients related to gastric cancer.

Num	Author/Year/Study	Country	Study Design	Population, *n* (%)	Cases vs. Control OR [95% CI]	
					Protective Factor	Risk Factor
1	De Stefani et al., (2000) [20]. Dietary carotenoids and risk of gastric cancer: a case–control study in Uruguay	Uruguay	Case- Control	Cases Males, 85 (70.8) Females, 35 (29.2) Control Males, 255 (70.8) Females, 105 (29.2)	Consumption of Vitamin C (>16,945 mg/d) 0.50 [0.30–0.99] α-carotene (>130 mg/d) 0.26 [0.17–0.65] Lycopene (>3448 mg/d) 0.24 [0.19–0.73]	NR
2	De Stefani et al., (2000) [20]. Plant Sterols and Risk of Stomach Cancer: A Case–Control Study in Uruguay	Uruguay	Case- Control	Cases Males, 85 (70.8) Females, 35 (29.2) Control Males, 255 (70.8) Females, 105 (29.2)	Consumption of β-Sitosterol (≥56.1 mg/d) 0.34 [0.15–0.76] Total phytosterols (≥82.6 mg/d) 0.33 [0.15–0.75]	NR
3	Muñoz et al., (2001) [12]. A case–control study of gastric cancer in Venezuela	Venezuela	Case- Control	Cases 292 (37.5) Control 485 (62.5)	Consumption of *^+^ Protein 0.56 [0.37–0.86] Polyunsaturated fat 0.63 [0.41–0.97] Niacin 0.62 [0.41–0.94]	NR
4	López-Carrillo et al., (2003) [21]. Capsaicin consumption, Helicobacter pylori positivity and gastric cancer in Mexico	Mexico	Case- Control	Cases Males, 133 (56.84) Females, 101 (43.16) Control Males, 266 (56.84) Females, 202 (43.16)	NR	Consumption of Capsaicin (30–89.9 mg/d) 1.60 [1.06–2.41]
5	Hernández-Ramírez et al., (2009) [22]. Dietary intake of polyphenols, nitrate and nitrite and gastric cancer risk in Mexico City	Mexico	Case- Control	Cases 248 (34.1) Control 478 (65.9)	Consumption of Cinnamic acids (>127.0 μg/d) 0.52 [0.34–0.81] Lignans (75.5 μg/d) 0.42 [0.27–0.65] Coumestrol (1.9 mg/d) 0.45 [0.29–0.70] Total nitrate (>141.7 mg/d) 0.61 [0.39–0.96] Nitrate in fruits and vegetables (>134.9 mg/d) 0.62 [0.40–0.97]	Consumption of Nitrate in animal (>3.9 mg/d) 1.92 [1.23–3.02] Nitrite in animal (>0.4 mg/d) 1.56 [1.02–2.40]
6	Trujillo-Rivera et al., (2018) [17]. Risk factors associated with gastric cancer in Mexico: education, breakfast, and chili	Mexico	Case- Control	Cases Males, 27 (58.7) Females, 19 (41.3) Control Males, 27 (58.7) Females, 19 (41.3)	NR	Consumption of Capsaicin (>29.9 mg/d) 3.00 [1.14–9.23]

OR, odds ratio; 95% CI, 95% confidence interval; NR, not reported; mg, milligrams; mg, micrograms; d, day; ^*^ OR from second tercile (Q2); ^+^ Consumption or frequency is not reported.

## Data Availability

The original contributions presented in the study are included in the article, further inquiries can be directed to the corresponding author.

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
