# Peer review of "Dietary Components Associated with the Risk of Gastric Cancer in the Latin American Population: A Systematic Review and Meta-Analysis"

_foods, 2025, doi:10.3390/foods14061052_

Round 1

Reviewer 1 Report

Comments and Suggestions for Authors

The manuscript by Avila-Nava et al.: Dietary components associated with the risk of gastric cancer in the Latin American population. A systematic review and meta-analysis

Gastric cancer is one of the most common cancer types worldwide and in general has a poor prognosis. Latin American countries are among the 20 countries with the highest incidence of gastric cancer. Thus, identifying the protective and risk factors associated with this disease is of importance and should be a mandatory task for health authorities. The rationale for this systematic review and meta-analysis in the context of existing knowledge is well described.

Using 13 articles in total, the authors concluded that the consumption of dairy fresh fruits and vegetables contributed to the decrease of gastric cancer; the regular consumption of fresh meat and eggs, salted, canned, and pickled food were associated as a risk factor for gastric cancer; the regular consumption of micro and macronutrients such as carotenoids, sterols, polyphenols, proteins, and polyunsaturated fatty acids showed a protective effect against gastric cancer, and that the regular consumption of capsaicin can increase the risk for gastric cancer in the Latin American population. Several limitations of this study are already stated in the Discussion section.

However, the authors did not define the „events“ out of „total“ number used for meta-analysis. Furthermore, the data synthesis suffers from serious flaws, including the repetition of data between data entries from single studies in the same meta-analysis (e.g. all vegetables and certain categories which are not independent). This repetition causes bias and inflates the results in terms of false robustness.

Fixed effect model seems adequate for data synthesis, but the apparent lack of heterogeneity is caused by repeating the results from single studies and from including too low number of data entries. Additionally, random effects model is more appropriate for biological and medical studies with representative populations.

Author Response

Response to Reviewer 1

Dear Reviewer, thank you for taking the time to assess this manuscript and provide us with your comments and suggestions. Please find below the answers to your comments and suggestions point by point.

Comments: The manuscript by Avila-Nava et al.: Dietary components associated with the risk of gastric cancer in the Latin American population. A systematic review and meta-analysis

Gastric cancer is one of the most common cancer types worldwide and in general has a poor prognosis. Latin American countries are among the 20 countries with the highest incidence of gastric cancer. Thus, identifying the protective and risk factors associated with this disease is of importance and should be a mandatory task for health authorities. The rationale for this systematic review and meta-analysis in the context of existing knowledge is well described.

Using 13 articles in total, the authors concluded that the consumption of dairy fresh fruits and vegetables contributed to the decrease of gastric cancer; the regular consumption of fresh meat and eggs, salted, canned, and pickled food were associated as a risk factor for gastric cancer; the regular consumption of micro and macronutrients such as carotenoids, sterols, polyphenols, proteins, and polyunsaturated fatty acids showed a protective effect against gastric cancer, and that the regular consumption of capsaicin can increase the risk for gastric cancer in the Latin American population. Several limitations of this study are already stated in the Discussion section.

Response: Thank you for your comments.

Comments: However, the authors did not define the „events“ out of „total“ number used for meta-analysis. Furthermore, the data synthesis suffers from serious flaws, including the repetition of data between data entries from single studies in the same meta-analysis (e.g. all vegetables and certain categories which are not independent). This repetition causes bias and inflates the results in terms of false robustness.

Response: Thank you for your comment. In the statistical analysis section, we specified that each study corresponds to the consumption of the different groups of fruits from the subjects (cases and controls), and N corresponds to the total subjects in each work. Hence, we added the following sentence in order to be clearer:

Forest plots were performed according to the consumption of the different groups of food for cases and controls with respect to the total number of subjects recruited in each study and group (N). The OR and 95% interval confidence (95% IC) were calculated for each study and the pooled group using Review Manager (RevMan) statistical software version 5.4 (The Cochrane Collaboration)”.

In addition, we recalculate the overall effect for each group of foods, considering no duplicate date entries for each study.

 Comments: Fixed effect model seems adequate for data synthesis, but the apparent lack of heterogeneity is caused by repeating the results from single studies and from including too low number of data entries. Additionally, the random effects model is more appropriate for biological and medical studies with representative populations.

Response: Thank you for your thorough revision. We recalculate the overall effects using the random effects model removing excess entries. Therefore, the heterogeneity is lower with this strategy.

Reviewer 2 Report

Comments and Suggestions for Authors

The review article Dietary components associated with the risk of gastric cancer in 2 the Latin American population. A systematic review and meta-3 analysis aims to determine the main dietary components associated 74 with gastric cancer present in the Latin American population. The topic of the study corresponds with the scope of the Journal.

The meta-analysis has identified 483 studies, and 13 were included in the systematic review, following PRISMA criteria.

The introduction is clear and consistent, emphasizing gastric cancer incidence as major concern in Latin America Population.

The Materials and Methods section provides detailed information for the contemporary methodology used for study design, data excretion, selection of studies, their quality assessment and statistical analysis.

The manuscript provides 45 references, as more than half of them are published in the last five years.

The Results are clearly presented, well organized and systematized in 5 Figures, along with supplementary files.

Discussion section provides interpretation of results and their comparison with other studies. However, the interpretation of results regarding nitrates and nitrites requires some refinements [L285-294]. Special revision of L287-290 “In this sense, fresh fruits and vegetables contain nitrosamines (may me nitrates) compounds and represent an important part of a healthy diet, because their consumption provides benefits, so they can be considered as protective factors against gastric cancer [33-35]” is needed.

Exposure to nitrite via nitrite-containing food such as processed meat is a minor part. In fact, the main source of exposure to nitrite is indirectly through the reduction of dietary nitrate from vegetables (e.g. lettuce, celery, spinach) [https://doi.org/10.1016/j.foodres.2023.112595].

Furthermore “nitrate and nitrite form animals” should be replaced with “nitrates and nitrites in meat products”. In this respect, L44-45 in Introduction section also needs correction.

Thanking into consideration the good quality of the work, the article should be accepted for publishing after minor corrections.

Author Response

Response to Reviewer 2

Dear Reviewer, thank you for taking the time to assess this manuscript and provide us with your comments and suggestions. Please find below the answers to your comments and suggestions point by point.

Comments: The review article Dietary components associated with the risk of gastric cancer in the Latin American population. A systematic review and meta-analysis aims to determine the main dietary components associated with gastric cancer present in the Latin American population. The topic of the study corresponds with the scope of the Journal.

The meta-analysis has identified 483 studies, and 13 were included in the systematic review, following PRISMA criteria.

The introduction is clear and consistent, emphasizing gastric cancer incidence as major concern in Latin America Population.

Response: Thank you for your comments.

Comments: The Materials and Methods section provides detailed information for the contemporary methodology used for study design, data excretion, selection of studies, their quality assessment and statistical analysis.

The manuscript provides 45 references, as more than half of them are published in the last five years.

The Results are clearly presented, well organized and systematized in 5 Figures, along with supplementary files.

Response: Thank you for your comments.

Comments: Discussion section provides interpretation of results and their comparison with other studies. However, the interpretation of results regarding nitrates and nitrites requires some refinements [L285-294]. Special revision of L287-290 “In this sense, fresh fruits and vegetables contain nitrosamines (may me nitrates) compounds and represent an important part of a healthy diet, because their consumption provides benefits, so they can be considered as protective factors against gastric cancer [33-35]” is needed.

Exposure to nitrite via nitrite-containing food such as processed meat is a minor part. In fact, the main source of exposure to nitrite is indirectly through the reduction of dietary nitrate from vegetables (e.g. lettuce, celery, spinach) [https://doi.org/10.1016/j.foodres.2023.112595].

Response: Thank you for your review and comments. We reorganized the discussion section according to your suggestions, and we added the following:

“It is important to note that the risk of cancer from nitrate consumption depends primarily on the sources from which these compounds are ingested, since nitrates are present in a wide variety of foods. In fact, between 80% and 85% of human exposure to nitrates comes from vegetables [35]; however, these foods are not linked to the development of gastric cancer because vegetables are also rich in vitamin C, vitamin E, polyphenols and fiber, compounds that reduce the formation of nitrosamines, which are identified as carcinogens [36]. On the other hand, processed meats lack reducing agents, allowing absorbed nitrate to be actively transported to the salivary glands, where oral bacteria reduce it to nitrite [37]. In the digestive tract, nitrites can be converted into nitrosating agents, which can lead to the nitrosation of biogenic amines and, consequently, the formation of N-nitrosamines [37, 38].

In addition, we added three new references, including the study conducted by Niklas, et al (2023).

 Comments: Furthermore “nitrate and nitrite form animals” should be replaced with “nitrates and nitrites in meat products”. In this respect, L44-45 in Introduction section also needs correction.

Response: we replaced the nitrates and nitrites in meat products in the Abstract (L44-45), and the Results: 3.9. Micronutrients (L276-277) according to your suggestions.

Comments: Thanking into consideration the good quality of the work, the article should be accepted for publishing after minor corrections.

Response: Thank you for your review.
